# Post-stroke fatigue: A factor associated with inability to return to work in patients <60 years—A 1-year follow-up

**Nicole Anna Rutkowski[1], Elham Sabri[2], Christine Yang[3]***

**1** Faculty of Social Sciences, School of Psychology, University of Ottawa, Ottawa, Canada, **2** Ottawa Hospital Research Institute, Ottawa, Canada, **3** Department of Physical Medicine and Rehabilitation, Elisabeth Bruyère Hospital, Ottawa, Canada

* cyang@bruyere.org

## Abstract

This study investigated the association between post-stroke fatigue and inability to return to work/drive in young patients aged <60 years with first stroke who were employed prior to infarct while controlling for stroke severity, age, extent of disability, cognitive function, and depression. The Fatigue Severity Scale (FSS) was used to evaluate post-stroke fatigue in this 1-year prospective cohort study. Follow-ups were completed at 3, 6, and 12 months post rehabilitation discharge. A total of 112 patients were recruited, 7 were excluded, due to loss to follow-up (n = 6) and being palliative (n = 1), resulting in 105 participants (71% male, average age 49 ±10.63 years). Stroke patients receiving both inpatient and outpatient rehabilitation were consecutively recruited. Persistent fatigue remained associated with inability to return to work when controlling for other factors at 3 months (adjusted OR = 18, 95% CI: 2.9, 110.3, p = 0.002), 6 months (adjusted OR = 29.81, 95% CI: 1.7, 532.8, p = 0.021), and 12 months (adjusted OR = 31.6, 95% CI: 1.8, 545.0, p = 0.018). No association was found between persistent fatigue and return to driving. Fatigue at admission was associated with inability to return to work at 3 months but not return to drive. Persistent fatigue was found to be associated with inability to resume work but not driving. It may be beneficial to routinely screen post-stroke fatigue in rehabilitation and educate stroke survivors and employers on the impacts of post-stroke fatigue on return to work.

## Introduction

Stroke continues to be a major contributor to mortality and disability rates worldwide. Globally, the incidence of ischemic stroke has risen 25% in working aged adults over the last 20 years [1]. Return to work has been identified as a key priority in the National Institute of Health and Care Excellence (NICE) guidelines on stroke rehabilitation [2] and identified as a recommendation in the Canadian Stroke Best Practice Guidelines [3]. Return to work outcomes post-stroke vary greatly ranging between 22–75% [4–6]. Additionally, the length of time post stroke and returning to work fluctuate, ranging from a few months to upwards of

**Data Availability Statement:** All relevant data are within the paper and its Supporting information files.

**Funding:** This study was funded by the Bruyère Academic Medical Organization Incentive fund. The

funds were awarded to CY. The funders had no role in study design, data collection and analysis, decision to publish, or preparation of the manuscript.

**Competing interests:** The authors have declared that no competing interests exist.

three years [7]. In regards to driving, 26.7% of stroke survivors reported returning to driving within one month, despite evidence-based guidelines recommending against it [3], and 83.8% had driven on at least one occasion within a year of their stroke [8].

Young stroke survivors face higher expectations for recovery, in addition to increased pressure to return to work due to financial necessity [9]. Returning to work may be dependent on returning to drive in order to commute to places of employment, which may contribute to individuals returning to driving earlier despite risks [8]. Return to work not only provides financial security and independence but successfully returning to work has been shown to increase life satisfaction, subjective well-being, and health-related quality of life [9–11]. However, patients frequently report substantial barriers to the resumption of work [12,13], with post-stroke fatigue being reported as the greatest barrier to returning to work within the first year after stroke [9].

Post-stroke fatigue has been defined as an overwhelming sense of tiredness or exhaustion, lack of energy, or difficulties sustaining routine actions in which rest is unrefreshing and there is a disturbed balance between motivation and effectiveness [14,15]. Fatigue has been found to be a time-dependent factor. It can arise as early as 10 days after stroke onset and can persist for more than three years post-stroke [16,17]. It is reported to affect between 38%-77% of stroke survivors [14,18–20]. Fatigue has been described to be an 'invisible' impairment that has a significant impact on functioning and that patients report feeling ill-equipped to handle [9,21]. Studies have found an association between post-stroke fatigue and a lower likelihood of returning to paid employment [22], as well as significant reductions in work load up to 2 years post stroke [9,23]. Regarding driving, young stroke survivors have been found to be less likely to return to driving up to 12 months post-stroke if they reported symptoms of depression or fatigue [8]. Further, fatigue at three months post-stroke may indirectly influence driving resumption through level of strength and motor activity [24].

In addition to post-stroke fatigue, patients experience severe residual impairments in physical, emotional, and cognitive functioning that may influence ability to return to work or drive [4,8]. Previous studies have reported that stroke severity, functional, cognitive, and emotional deficits may be barriers to the resumption of work and driving [8,25]. Depression and post-stroke fatigue can be difficult to distinguish, as fatigue is a common feature of depression [26]. There is some evidence to support depression as a predictor of increased fatigue in younger stroke survivors [19,27]. However, studies have demonstrated that fatigue can develop in the absence of depression or a significant cognitive impairment [14,17].

The primary objective of this study was to assess the effect of persistent fatigue on inability to return to work (part or full time) and drive in working-age stroke survivors <60 years at three measurement points: 3 months, 6 months, and 12 months post discharge from rehabilitation while controlling for confounders factors like severity of stroke, age, extent of disability, cognitive function, and depression. The secondary objective was to assess the effect of fatigue on admission to predict inability to return to work and drive at 3, 6, and 12 months. We hypothesized that post-stroke fatigue would be strongly associated with inability to return to work and drive. To our knowledge, this is the first study to investigate young stroke survivors return to work and drive while controlling for the abovementioned confounders.

## Methods

This study was a longitudinal, observational study conducted at Élisabeth Bruyère Hospital in Ottawa between 2014–2016. Élisabeth Bruyère Hospital is the tertiary deliverer of stroke rehabilitation in the Ottawa region. The average transfer time from acute care to rehabilitation was 20 days. All patients were assessed at admission. Patients were followed up to 12 months post

discharge from rehabilitation, however patients were only assessed until they returned to work as per usual practice, therefore not every patient was followed for the entirety of the year resulting in a reduced sample size over time. Stroke patients receiving both inpatient and outpatient rehabilitation at the same institution were consecutively recruited for one year. Inpatients received between 4 to 7 weeks of rehabilitation and outpatients received between 4 and 10 weeks. The inclusion criteria were: 1) age 18–60 years, 2) first-ever stroke, 3) ischemic or hemorrhagic stroke, 4) employed either full time or part time at the time of the stroke, and 5) able to provide informed consent. The exclusion criteria were: 1) severe aphasia with Boston Diagnostic Aphasia Examination scale < 4, 2) recurrent stroke or major medical illness since the time of initial admission to stroke rehabilitation, and 3) severe cognitive impairment such as inability to answer questions or undergo a reliable interview. This study received ethics approval from Bruyère Research Ethics Board (Bruyère REB Protocol M16-13-016). Patients were screened based on study criteria by the attending physician; once identified, a nurse obtained verbal consent for a research assistant to approach. The research assistant explained the study and collected written informed consent. A total of 112 patients were recruited, 7 were excluded, due to loss to follow-up (n = 6) and being palliative (n = 1). All eligible patients were asked to participate and 100% of patients agreed to be part of the study.

## Outcome measures

The Fatigue Severity Scale (FSS) was used to evaluate post-stroke fatigue [16,28] and was assessed by the attending physician at admission to rehabilitation and at 3, 6, and 12 months post discharge. The FSS measures how a person's level of fatigue interferes with daily life. It consists of nine items on a 7-point Likert scale ranging from "strongly disagree" to "strongly agree". The scale has demonstrated a high internal consistency of 0.89 [29]. Patients with an FSS score ≥4.0 were considered fatigued. This cut-off value was chosen as less than 5% of healthy controls rate their fatigue at ≥4.0, whereas 69–90% of patients with medical disorders (e.g., multiple sclerosis, myocardial infarction, stroke) experience fatigue at or above the cut-off level [28,30].

Return to work and drive were captured by the attending physician during patients' standard follow-up appointments at 3, 6, and 12 months post discharge or until the patient reported returning to work. Return to work was defined as either return to full-time hours (i.e., without reduced hours or modified position duties) or on a modified return to work plan (i.e., reduced hours and/or modified duties). In Canada when patients with stroke engage in rehabilitation, return to work planning involves the same employment they held prior to the stroke onset, therefore all the stroke patients in this study who returned to work would have returned to their previous employment on a modified capacity or full-time. Fatigue at admission and persistent fatigue at 3, 6, and 12 months were used as predictors for return to work and drive. Return to drive was determined by following criteria for fitness to drive based on best practice guidelines and the ministry of transportation (e.g., medical risk management, physical, cognitive, and visual abilities). The attending physician inquired whether patients had returned to work or drive during the follow-up assessments. When a patient returned to work, collecting independent predictor variables ceased for them.

## Confounders

Clinically relevant confounders were selected based on previously identified post-stroke fatigue related factors and factors that impact return to work in stroke survivors [4,25,27]. Stroke severity, extent of disability, cognitive function, and depression were assessed on

admission and at 3, 6, and 12 months post discharge using the following outcomes. All data was gathered from clinical documentation.

Functional Independence Measure (FIM) is an 18-item tool that assesses a patient's level of disability, burden of care, and monitors improvement in function during rehabilitation. It captures bowel and bladder control, transfers, locomotion, communication, social cognition as well as other self-care activities. It has demonstrated high inter-rater reliability of 0.86–0.88 [31]. The FIM was assessed by the rehabilitation team within 72 hours of admission to rehabilitation.

National Institutes of Health Stroke Scale (NIHSS) is commonly used to measure stroke severity in patients. It evaluates multiple levels of physical functioning (e.g., motor movements, speech, vision, and sensory deficits). It has been shown to be a significant predictor of stroke patient outcomes [32]. The NIHSS was assessed by the acute care team at time of stroke onset.

Modified Rankin Disability Scale (mRS) is a widely used clinician-reported tool that measures the degree of disability and has strong test re-test reliability of 0.81–0.95. The scale is scored from 0–6, ranging from no symptoms (0) to death (6). Scores 1–5 denote slight, moderate, or severe disability [33]. The mRS was assessed by the attending physician.

Montreal Cognitive Assessment (MoCA) is a 30-point test that assesses cognitive impairment in several cognitive domains (i.e., short-term memory, attention, concentration, and language). It is scored from 0–30, with scores of 26 or higher being considered normal. It has demonstrated a high internal reliability of 0.78 [34,35]. The MoCA was assessed by an occupational therapist.

Beck Depression Inventory II (BDI-II) is a 21-item self-report inventory that assesses the severity of depressive symptoms. Items are scored on scales of 0–3 with total scores ranging from 0–63, with higher scores indicated greater severity of depression. The scale has shown high internal consistency with outpatients ($\alpha = 0.92$) [36]. BDI-II was administered by a trained research assistant by means of a face-to-face interview.

## Statistical analysis

The data were analyzed using SAS V 9.4 software (SAS institute, Cary, NC, USA). All p-values reported are two-sided with the level of significance set at <0.05. Baseline characteristic of patients were reported using Means (± standard deviation) for continuous and percentages for categorical variables. Fatigue was used as a binary variable with FSS ≥ 4.0 indicating that. Numbers and percentages of stroke survivors not returned to work at 3, 6, and 12 months were reported for the two groups of fatigued and non-fatigued survivors at each time point. For the data analysis, survivors were presumed to have remained at work or drive at 6 or 12 months if they had indicated they had returned to work/drive at an earlier time point. Univariable logistic regression models with maximum likelihood estimates were used to assess the effect of fatigue as well as other independent variables including age, FIM, NIHSS, mRS, MoCA, and BDI at 3, 6, and 12 months for inability to return to work concurrently with dependent variables in separate models. Unadjusted Odds Ratios with 95% confidence intervals were calculated to represent the effect of fatigue.

Multivariable logistic regression models were used to control for the effect of other known confounders present at each time point and adjusted Odds Ratios with 95% confidence intervals and p-values were reported. The variables included in the multivariable models were age, fatigue, severity of stroke, extent of disability, cognitive function, and depression. Multicollinearity in all multivariable models were assessed by looking at the lowest tolerance value and highest value for variance inflation. Similar analysis was used to assess the effect of fatigue on inability to return to drive.

Univariable logistic regression models with maximum likelihood estimates were also used to assess the effect of fatigue at admission as well as all confounders on inability to return to work at 3, 6, and 12 months as dependent variables in separate models. A multivariable logistic regression models to control for effect of other known confounders measured at admission was also calculated.

## Results

A total of 112 patients were recruited and 105 patients were followed for one-year or until they returned to work before the one-year mark. The mean age of participants was 49 ±10.63 years. Participants were predominately male (71.4%), married (70.5%), working full time prior to stroke (92.4%), who had suffered an ischemic stroke (87.6%). Mean fatigue severity at admission was 4.22 ±1.55. On admission, 67% of patients reported fatigue above the cut-off. See Table 1 for patient demographics.

### Return to work

By three months 63 out of 105 (60%) of survivors did not return to work. After six months, 46/ 105 (43.8%) of participants had not return to their occupation. By one year, 34/104 (32.7%) had not resumed employment, indicating 67.3% of our sample had returned to work one-year post-rehabilitation.

Patients who continued to report fatigue at 3 months were less likely to return to work than patients whose fatigue resolved (unadjusted OR = 41.6, 95% CI: 11.2, 154.1, p = < .001). Further, fatigue also had an impact at 6 months (unadjusted OR = 40.6, 95% CI: 4.9, 338.3, p = 0.001). At 12 months, the effect was observed even though the statistical significance was not maintained (unadjusted OR = 4.3, 95% CI: 0.96, 19.2, p = 0.056). The sample size to assess the effect of persistent fatigue reduced to 63 and 45 patients at 6 and 12 months respectively based on number of patients who had not returned to work. Multivariable logistic regression was used to adjust for the effect of confounders including age, FIM and NIHSS on admission, mRS, MoCA, and BDI measured at time of outcome assessment. Presence of fatigue remained a statistically significant risk factor for inability to return to work at 3 months (adjusted OR = 18, 95% CI: 2.9, 110.3, p = 0.002), at 6 months (adjusted OR = 29.8, 95% CI: 1.7, 532.8, p = 0.021), and at 12 months (adjusted OR = 31.6, 95% CI: 1.8, 550, p = 0.018), when controlled for other factors.

Fatigue at admission was a predictor of inability to return to work at 3 months, (unadjusted OR = 6.9, 95% CI: 2.9, 17.6, p = <0.0001) and 6 months (unadjusted OR = 4.3, 95% CI: 1.7, 10.7, p = 0.002), suggesting that patients who report fatigue on admission may be less likely to return to work by 6 months compared to patients who do not report fatigue. The association between fatigue at admission and inability to return to work at 12 months showed a similar trend but was not statistically significant (unadjusted OR = 2.6, 95% CI: 0.99, 6.7, p = 0.054). However, these results were only maintained at 3 months (adjusted OR = 5.7, 95% CI: 1.5, 23.0, p = 0.012) when adjusted for confounders. These findings are presented in Table 2. Univariable analysis was used to explore the importance of each confounding factor including age, fatigue, severity of stroke, extent of disability, cognitive function, and depression. No other factors besides fatigue and disability (mRS) were significantly associated with inability to return to work, but disability lost significance at 12 months. When controlling for the other confounders, disability only remained significant at 3 months. In multivariable models for inability to return to work, multicollinearity was examined through the Variance Inflation Factor and Tolerance. The highest value for Variance Inflation Factor (2.55) and lowest value for Tolerance (0.39) indicated a lack of multicollinearity.

**Table 1. Demographic and stroke characteristics.**

|  |  | Mean (SD) or n (%) |
|---|---|---|
| **Age** |  | 49 (10.63) |
| **Sex** | Male | 75 (71%) |
|  | Female | 30 (28%) |
| **Marriage status** | Married | 74 (71%) |
|  | Single | 17 (16%) |
|  | Divorced | 13 (12%) |
|  | Widowed | 1 (1%) |
| **Working status** | Working full time | 97 (92%) |
|  | Working part time | 8 (8%) |
| **Working type** | Physical | 16 (15%) |
|  | Clerical or technical | 30 (28%) |
|  | High professional | 59 (57%) |
| **Driving status** | Driver | 97 (92%) |
|  | Non-Driver | 8 (8%) |
| **Stroke type** | Ischemia | 91 (87%) |
|  | Hemorrhage | 14 (13%) |
| **Side of stroke** | Right | 58 (55%) |
|  | Left | 33 (32%) |
|  | Bilateral | 14 (13%) |
| **Stroke location** | Cortical | 42 (40%) |
|  | Subcortical | 37 (35%) |
|  | Cerebellum | 12 (12%) |
|  | Brainstem | 14 (13%) |
| **Stroke deficits** | Hemiparesis | 85 (81%) |
|  | Dysarthria | 18 (17%) |
|  | Apraxia | 7 (7%) |
|  | Neglect | 5 (5%) |
|  | Ataxia | 8 (8%) |
| **Mean Scores on Admission (N = 105)** |  |  |
| **FIM** |  | 88.82 (20.58) |
| **NIHSS** |  | 5.3 (4.74) |
| **FSS** |  | 4.22 (1.55) |
| **MoCA** |  | 24.27 (3.80) |
| **BDI** |  | 11.31 (8.57) |
| **mRS** |  | 2.24 (1.01) |

**Note.** FIM: Functional Independence Measure, NIHSS: National Institutes of Health Stroke Scale, FSS: Fatigue Severity Scale, MoCA: Montreal Cognitive Assessment Score, BDI: Beck Depression Inventory, mRS: Modified Rankin Disability Scale.

## Return to drive

On the other hand, less than half 41/97 (42.3%) of participants did not return to drive after three months. The minority 24/97 (24.7%) of participants had not to resumed driving by 6 months and only 20/96 (20.8%) have not returned to driving by one year.

As seen in Table 3, presence of fatigue at 3 months suggests a lower likelihood of returning to driving (unadjusted OR = 6.8, 95% CI: 2.8, 16.8, p = < .001), however these effects were not maintained when controlling for confounders (adjusted OR = 1.4, 95% CI: 0.3, 6.6, p = 0.65).

**Table 2. Association between fatigue and cofounders and not returning to work.**

| | | Fatigue Yes | Fatigue No | Unadjusted Odds Ratio (95% CI) | p-value | Adjusted Odds Ratio (95% CI)* | p-value |
|---|---|---|---|---|---|---|---|
| **A. Persistent Fatigue measured at time of outcome** | | | | | | | |
| Not-Return to work N (%) | 3M | 48/51 (94.1) | 15/54 (27.8) | 41.6 (11.2, 154.1) n = 105 | < .0001 | 18.0 (2.9, 110.3) n = 97 | **0.0018** |
| | 6M | 33/34 (97.1) | 13/29 (44.8) | 40.6 (4.9, 338.3) n = 63 | **0.0006** | 29.8 (1.7, 532.8) n = 58 | **0.021** |
| | 12M | 21/24 (87.5) | 13/21 (61.9) | 4.3 (0.96, 19.2) n = 45 | 0.0558 | 31.56 (1.5, 549.9) n = 43 | **0.018** |
| **B. Fatigue on Admission** | | | | | | | |
| Not-Return to work N (%) | 3M | 52/69 (75.4) | 11/36 (30.6) | 6.9 (2.8, 17.0) n = 105 | < .0001 | 5.8 (1.5, 23.0) n = 97 | **0.0116** |
| | 6M | 38/69 (55.1) | 8/36 (22.2) | 4.3 (1.7, 10.7) n = 105 | **0.0019** | 2.3 (0.68, 7.5) n = 97 | 0.1823 |
| | 12M | 27/69 (39.1) | 7/35? (20.0) | 2.6 (0.99, 6.7) n = 104[1] | 0.0536 | 1.06 (0.31, 3.7) n = 97 | 0.9218 |
| **C. Cofounders measured at time of outcome** | | | | | | | |
| FIM | 3M | | | 0.98 (0.95, 0.100) | 0.017 | 0.99 (0.99,1.05) | 0.769 |
| | 6M | | | 0.99 (0.97, 1.02) | 0.492 | 0.98 (0.93, 1.04) | 0.585 |
| | 12M | | | 0.96 (0.92, 0.100) | 0.029 | 0.97 (0.90, 1.03) | 0.34 |
| mRS | 3M | | | 13.7 (5.1, 36.8) | <0.0001 | 7.59 (1.7, 33.9) | **0.008** |
| | 6M | | | 34.64 (4.15, 289.33) | 0.001 | 7.26 (0.58, 91.69) | 0.125 |
| | 12M | | | 8.1 (1.59, 1.11) | 0.192 | 16.87 (0.69, 414.47) | 0.08 |
| MoCA | 3M | | | 0.73 (0.6, 0.88) | 0.001 | 0.88 (0.62, 1.23) | 0.448 |
| | 6M | | | 0.77 (0.59, 1.02) | 0.068 | 0.73 (0.42, 1.25) | 0.246 |
| | 12M | | | 0.81 (0.59, 1.11) | 0.192 | 1.48 (0.86, 2.55) | 0.155 |
| BDI | 3M | | | 1.2 (1.11, 1.3) | <0.001 | 1.05 (0.95, 1.16) | 0.336 |
| | 6M | | | 1.26 (1.09, 1.46) | 0.002 | 1.29 (1.01, 1.65) | **0.046** |
| | 12M | | | 0.98 (0.92, 1.05) | 0.590 | 0.99 (0.87, 1.13) | 0.855 |
| NIHSS | 3M | | | 1.32 (1.12, 1.55) | 0.001 | 1.21 (0.97, 1.52) | 0.096 |
| | 6M | | | 0.98 (0.88, 1.1) | 0.776 | 0.98 (0.80, 1.21) | 0.87 |
| | 12M | | | 1.34 (1.01, 1.77) | 0.041 | 1.47 (0.92, 2.35) | 0.104 |
| Age | 3M | | | 0.998 (0.96, 1.04) | 0.919 | 1 (0.93, 1.08) | 0.981 |
| | 6M | | | 1.02 (0.96, 1.08) | 0.609 | 1.07 (0.9, 1.26) | 0.46 |
| | 12M | | | 1.05 (0.97, 1.12) | 0.221 | 1.17 (0.97, 1.40) | 0.094 |

* In A: Effect of persistent fatigue was controlled for FIM and NIHSS on admission, and mRS, MoCA, and BDI measured at time of outcome assessed.

* In B: Effect of fatigue at baseline was controlled for FIM, mRS, MoCA, BDI, and NIHSS measured at admission.

*In C: All confounders were analyzed in univariable and multivariable models.

[1]One patient was lost to follow up after 6 months.

Presence of fatigue at 6 and 12 months was not found to be associated with inability to return to driving, suggesting stroke survivors may be resuming driving more rapidly than work despite continuing to experience fatigue. Fatigue on admission suggests a higher likelihood of not returning to drive at 3 (unadjusted OR = 2.9, 95% CI: 1.2, 7.1, p = 0.023) and 6 months (unadjusted OR = 3.5, 95% CI: 1.1, 11.2, p = 0.036), however these effects did not remain after controlling for cofounders.

## Discussion

This study monitored the impact of persistent fatigue on resumption of work in a group of 105 stroke patients, controlling for several potential confounding factors–severity of stroke, age, extent of disability, cognition, and post-stroke depression. These findings suggest that post-stroke fatigue may be an important factor for inability to return to work regardless of stroke severity, age, extent of disability, cognitive function, and depression. Fatigue on admission suggested a lower likelihood of returning to drive at 3 and 6 months, however persistence of

**Table 3. Association between fatigue and not returning to drive.**

| | | Fatigue Yes | Fatigue No | Unadjusted Odds Ratio (95% CI) | P-value | Adjusted Odds Ratio (95% CI)* | P-value |
|---|---|---|---|---|---|---|---|
| **A. Persistent Fatigue measured at time of outcome** | | | | | | | |
| Not-Return to Drive N (%) | 3M | 30/46 (65.2) | 11/51 (21.6) | 6.8 (2.8, 16.8) n = 97 | < .0001 | 1.4 (0.3, 6.6) n = 91 | 0.646 |
| | 6M | 15/23 (65.2) | 9/18 (50.0) | 1.9 (0.5, 6.6) n = 41 | 0.329 | 0.6 (0.1, 6.4) n = 39 | 0.681 |
| | 12M | 10/12 (83.3) | 10/11 (90.9) | 0.5 (0.04, 6.4) n = 23 | 0.595 | The Maximum likelihood estimate may not exist. n = 22 | |
| **B. Fatigue on Admission** | | | | | | | |
| Not-Return to Drive N (%) | 3M | 32/63 (50.8) | 9/34 (26.5) | 2.9 (1.2, 7.1) n = 97 | **0.023** | 1.5 (0.4, 5.3) n = 91 | 0.486 |
| | 6M | 20/63 (31.7) | 4/34 (11.8) | 3.5 (1.1, 11.2) n = 97 | **0.036** | 1.3 (0.3, 5.8) n = 91 | 0.699 |
| | 12M | 16/62 (25.8) | 4/34 (11.8) | 2.5 (0.8, 8.3) n = 96[1] | 0.105 | 0.8 (0.1, 4.1) n = 90 | 0.766 |

* In A: Effect of persistent fatigue was controlled for FIM and NIHSS on admission, and mRS, MoCA, and BDI measured at time of outcome assessed.

* In B: Effect of fatigue at baseline was controlled for FIM, mRS, MoCA, BDI, and NIHSS measured at admission.

[1] One patient who did not return to drive by 6 months was missing the return to drive data at 12 months.

fatigue at 3, 6, and 12-months post discharge did not suggest an impact on return to driving after controlling for confounders.

## Return to work

Our findings are in accordance with a previous study investigating post-stroke fatigue as an important factor in inability to resume paid work following stroke while controlling for anti-depressant usage [22]. Building on the findings of this study, we additionally controlled for other potential confounders such as severity of stroke, age, extent of disability, and cognitive function. After controlling for these confounders, persistent fatigue continued to demonstrate an association with inability to return to work up to a year post-discharge.

Compared to previous studies, 67.3% of our sample had returned to work either full time or part-time one-year post rehabilitation, which falls on the higher end of the spectrum for expected return to work rates post-stroke [5,22]. However, it is not known whether partici-pants' return to work was successful as returning to work has been shown to intensify residual stroke impairments like fatigue, resulting in an inability to sustain employment long-term [13]. A large portion of our sample (85%) were previously employed in clerical or high profes-sional positions which may contribute to the high percentage returning to work, as patients in high professional or clerical positions have been found to be more likely to return to work [37], particularly to jobs that are less physically and psychologically demanding post-stroke [38]. The large sample of clerical or high professionals may be attributional to the nature of the city of Ottawa, Canada. As the capital of Canada, the work sector consists primarily of white-collar jobs in high technology and federal public service.

Post-stroke fatigue and depression can be difficult to distinguish [39], given that the two conditions have features in common and that fatigue is one of the criteria for diagnosis of a major depressive disorder in the DSM-5 [26]. In our study, after controlling for depression, fatigue continued to demonstrate predictive ability for not returning to work. Depression was found to be correlated with fatigue in our study as has been previous documented [40], how-ever fatigue was also present in non-depressed stroke patients. A study by Hackett et al. [5] did

not find an association between early depression and inability to return to work in stroke survivors. Our findings suggest that fatigue regardless of depression level may be a barrier to the resumption of work.

Further, fatigue remained a factor for not returning to work after adjusting for severity of stroke. A study by Hartke and Trierweiler [9] found patient's self-reported physical impairment as the most frequent barrier to returning to work. It is likely that once physical deficits decrease or patients become more accustomed to their post-stroke mobility, fatigue becomes the substantial barrier. Previous studies have found patients with good physical recovery to be the most disabled by fatigue and tend to rate fatigue as a more severe symptom [14,15]. Evidence also suggests that fatigue is more prevalent in younger than older stroke survivors [41], making fatigue a potentially important and neglected predictor for inability to return to work. Cognitive and neurological impairments have also been found to be associated with an inability to return to work [25]. In our study, while controlling for cognitive and neurological impairments, persistent fatigue continued to demonstrate predictive value up to 12 months.

## Return to driving

Driving is a challenging task which requires high cognitive, attentional, and motor demands. Fatigue and stroke severity may indirectly influence driving resumption through the level of strength and motor activity [24]. Interestingly, our study results showed fatigue did not extensively delay return to driving with or without controlling for confounders of physical limitations and cognitive impairments which are essential requirements for driving fitness. More than half of participants returned to driving within 3 months. A possible explanation for this may be that patients adapt by limiting duration of driving, modifying vehicles, or reducing the amount of time behind the wheel [21]. Further, driving may be a necessity in a country like Canada, where the geographic landscape is greatly dispersed. In addition, it may be a necessary step to returning to work [8]. This may add pressure on patients to return quickly to driving, despite continuing to experience persistent fatigue.

## Clinical and policy implications

Returning to work is an important component of community re-integration for young stroke survivors [2,12]. Frustration and a lack of knowledge around fatigue has been expressed by patients [21]. In general, patients report not receiving adequate information or vocational counselling on returning to work [21,42,43]. This may be particularly distressing, as the unexpected loss of income creates financial pressure, which has been found to be a driving force in stroke patients' desire to return to work [42]. Based on our clinical experience in Canada, additional pressure by insurance companies to return to work increases six months post-stroke. Further, disability claims where fatigue is the primary reason for inability to return to work on the Attending Physician's Statement are often denied. Despite evidence-based guidelines recommending routine screening and education of post-stroke fatigue, these recommendations to our knowledge have not been widely adapted into practice [44]. A recent update of the Canadian Stroke Best Practice Guidelines added more evidence for post-stroke fatigue [45]. It recommended that people who have experienced a stroke should be periodically screened for post-stroke fatigue during follow-up visits [45]. Our findings support that it is important to routinely screen for fatigue and implement policy changes that recognize fatigue as an important consideration for return to work when considering disability benefits.

Many individuals continue to experience persistent fatigue alongside cognitive and physical losses that make full-time employment unsustainable. Some studies have recommended

encouraging stroke survivors in flexible return to work planning, such as phased return to work and flexible working hours [43,46] yet dependent on the country these efforts may be hindered by employers due to a lack of governmental policies or laws. In Canada for example, medical professionals play an important role in advocating for workplace accommodations and duties for patients and the recommendations put forward are often respected by insurers and employers. However, in our experience greater resistance is encountered when advocating for patients with persistent fatigue. In order to encourage more successful return to work, it may be necessary to promote a dialogue between health care providers, insurance representatives, and employers to foster understanding and to facilitate more appropriate work accommodations for stroke survivors with persistent fatigue [38,42]. Particularly as a recent study by Ghoshchi and colleagues found that stroke survivors reported higher quality of life when they returned to the same employment and working hours they held prior to their stroke [47]. Workplace accommodations, especially work from home, may be more accessible to stroke survivors presently as many institutions have been compelled to develop infrastructure to support remote work given the global pandemic. However, a significant barrier for stroke survivors returning to work to physically demanding jobs remains [48]. Individualized return to work programs have been found to increase the likelihood three-fold of return to work among stroke survivors in South Africa by providing personalized accommodations and assessment of workplace challenges, though they require significant human resources they may be a worthwhile investment given the substantial economic burden of lost productivity among working aged stroke survivors [48,49].

It is necessary to advocate for increased education, adequate disability coverage, and awareness around the impact fatigue has on returning to work in order to create realistic expectations for employers, insurers, and stroke survivors [13]. Particularly, as studies have shown it can take upwards of three years post-stroke for survivors to return to work [7]. With high rates of fatigue experienced by stroke survivors, health professionals should prepare and educate patients about post-stroke fatigue and its vocational challenges, and counsel on management of post-stroke fatigue (i.e., energy conservation strategies, mindfulness based stress reduction, establishment of good sleep hygiene behaviors, and graduated exercise schedules) [50,51]. Neurorehabilitation programs such as functional electrical stimulation may be beneficial for return to work by improving walking, muscle weakness, and reducing fatigue through increased physical activity, however they may not be widely accessible [52]. Further research on neurorehabilitation programs, such as virtual reality and robot technology and whether enhanced motor learning and cognitive function is transferable to successful return to work is needed.

## Strengths and limitations

The strength of this study is its examination of persistent fatigue's effect on inability to return to work and drive over time in stroke survivors while controlling for several important confounders. There are limitations to this study, however. Firstly, post-stroke fatigue is difficult to measure due to its multiple components (i.e., cognitive, physical, and psychological) and is subject to the inherent bias of self-reporting. Additionally, fatigue may be impacted by comorbid medical conditions which were not assessed for in this study. Secondly, the generalizability of this study may be limited due to convenience sampling, which resulted in patients who were predominately male and in clerical or high professional positions. As previously mentioned, patients in high professional or clerical positions are more likely to return to work [37]. Further, patients with a recurrent stroke were excluded due to complexities in evaluating fatigue as an impact from stroke in this demographic. Existing fatigue may be present in patients with

a recurring stroke and these results may not be generalizable to this group. Thirdly, the standard process to return to work involves gradual return to modified hours and duties to previous employment, however this study did not collect information on the specific reduction of working hours. Longer follow-up of two or more years could have provided valuable information, particularly if employment was sustained or adjusted. As follow-up was ceased upon return to work, not all patients were followed for the entirety of the year, it is therefore unclear if their return to work was successfully sustained, or if those who had not resumed work would eventually return to work. Further, the results of this study may not be generalizable to all stroke survivors given our sample consisted primarily of white-collar professionals. Lastly, in any of the multivariable models, the tolerance value and variance inflation factor did not indicate a multicollinearity problem in the models, however in models that studied the effect of fatigue present at 6 or 12 months on inability to return to work or drive, the results should be interpreted with caution as the number of participants reduced significantly in these models. Therefore, these results may need to be replicated with a larger sample size. Future studies should explore the association between post-stroke fatigue and return to work in the long-term.

## Conclusions

Fatigue is an invisible impairment that may make returning to work more difficult for some stroke survivors and may require greater attention among stroke survivors, clinicians, insurance companies, and employers. This study found that persistent fatigue was highly associated with inability to return to work up to 12 months post discharge from rehabilitation, regardless of severity of stroke, age, cognitive impairment, or depression. However, post-stroke fatigue did not significantly affect ability to return to drive. It may be beneficial to comprehensively incorporated assessment of post-stroke fatigue into routine rehabilitation and educate stroke survivors, employers, and insurers about the impact of persistent fatigue on return to work.

## Supporting information

**S1 Dataset. Deidentified outcome and demographic data from study sample.**
(XLSX)

**S1 Data. Outcome and demographic coding sheet for data analysis.**
(DOCX)

## Acknowledgments

The authors would like to thank Dr. Hillel Finestone for editing this manuscript and the Ottawa Methods Centre for conducting the statistical analysis.

## Author Contributions

**Conceptualization:** Christine Yang.

**Data curation:** Nicole Anna Rutkowski, Elham Sabri.

**Formal analysis:** Elham Sabri.

**Funding acquisition:** Christine Yang.

**Methodology:** Christine Yang.

**Project administration:** Nicole Anna Rutkowski.

**Supervision:** Christine Yang.

**Writing – original draft:** Nicole Anna Rutkowski.

**Writing – review & editing:** Nicole Anna Rutkowski, Elham Sabri, Christine Yang.

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
