## [Decision Letter · Decision Letter 0]

21 May 2021

PONE-D-21-12930

Post-stroke fatigue: A factor associated with inability to return to work in patients <60 years - a 1-year follow-up.

PLOS ONE

Dear Dr. Yang,

Thank you for submitting your manuscript to PLOS ONE. After careful consideration, we feel that it has merit but does not fully meet PLOS ONE’s publication criteria as it currently stands. Therefore, we invite you to submit a revised version of the manuscript that addresses the points raised during the review process.

We look forward to receiving your revised manuscript.

Kind regards,

Simone Reppermund, PhD

Academic Editor

PLOS ONE

Journal Requirements:

Reviewers' comments:

Reviewer's Responses to Questions

**Comments to the Author**

1. Is the manuscript technically sound, and do the data support the conclusions?

Reviewer #1: No

Reviewer #2: Yes

2. Has the statistical analysis been performed appropriately and rigorously? 

Reviewer #1: No

Reviewer #2: Yes

3. Have the authors made all data underlying the findings in their manuscript fully available?

Reviewer #1: Yes

Reviewer #2: Yes

4. Is the manuscript presented in an intelligible fashion and written in standard English?

Reviewer #1: Yes

Reviewer #2: Yes

5. Review Comments to the Author

Reviewer #1: This is a single center study of 105 patients ages 18-60 who were employed prior to their first stroke and were followed for 12 months after to determine success at driving and return to work. Investigators evaluated the impact of fatigue using the Fatigue Severity Scale (FSS). They considered a score of greater than 4 significant fatigue. They found that individuals who were fatigued at each time point were less likely to return to work but that this did not have significant impact on driving. They also found that early fatigue was independently associated with not returning to work later.

This is an important topic. Post-stroke fatigue is more common than previously realized and a problem for individuals reintegrating into their prior home and workplace environments.

Generalizability- This is a relatively small group of patients with a single center. They are predominantly male and quite young.

Confounding factors and clinical significance- The authors do not account for many other potentially confounding causes- though do adjust for co-existing depression (using the Beck inventory), the patient's functional status (using the FIM, NIH stroke scale, and modified Rankin), and cognition (using the Moca). Other studies have found that when people remain fatigued chronically after stroke it is often due to other medical conditions, yet this is not evaluated and may be another reason why return to work is not possible. Age (if nearer retirement for example), and type of work may also be important. While I appreciate that fatigue was “independently associated”, a univariate analysis exploring other important factors AND including the relative weights of all important factors within their multivariable model (where they now just report the adjusted OR for fatigue) would be helpful to the reader to gauge the relative importance of each factor.

The abstract could be improved by being clear that they are focused on younger individuals with first stroke who were employed prior to infarct.

Within the Discussion, it is really too strong a statement to say that fatigue is the major driver for not returning to work, without some of the above analysis. It certainly is a contributing factor, but may not be the only, or even dominant one.

Reviewer #2: The authors should clarify why 89% of their patients were in Clerical or technical or High professional positions. It is a very high percentage and it implies a sampling bias.

In the abstract, please report the p-values of the odds ratios

I would suggest to use the First people language, referring to people/patients with stroke, and not to stroke survivors.

The percentages of RWT reported in Introduction are those related only to 2 studies (4,5), despite authors reported that this rate highly varies. Probably it is better to report a range of percentages obtained comparing more studies (such as well done for fatigue post stroke)

The OBJECTIVES paragraph should be merged with the paragraph at the end of Introduction (without any subtitle). It is quite strange to find the objective in methods section.

Is it possible to write the following sentence in a more clear manner: “For assessing the effect of fatigue on admission on return to work/drive, patients were assumed to have returned to work or drive if they had returned to work/drive at an earlier time point.”?

I would suggest to report OR results as follows: unadjusted OR=41.6, 95% CI: 11.2-154.1, p=<0.001, I suggest to avoid to report p<.000

When the OR was adjusted for confounders, were the confounders measured as fatigue at admission and also at the time of outcome, separately for the two analyses reported in the table?

Authors investigated the return to work, without dividing between the return to the same mansion or to the same working time (as done elsewhere, for example in “Return to Work and Quality of Life after Stroke in Italy: A Study on the Efficacy of Technologically Assisted Neurorehabilitation.”). It is a point to discuss, also because authors discussed that the time of driving could be reduced with respect to before stroke, the same could happen for return to work, despite not enough to allow this reduction.

Another fundamental aspect is the possibility or not to adapt the workplace/office and or the possibility to use or not special devices/aids for working. Authors should discuss also this aspect as done elsewhere (for example in “The effect of a workplace intervention programme on return to work after stroke: a randomised controlled trial”)

The Authors suggested “to support and encourage patients in flexible return to work planning”, however this aspect is often decided by companies and regulated by government laws. Authors could provide suggestions for promoting laws favouring this approach, not just a suggestion to give to patients.

The Discussion would also benefit if authors report also some neurorehabilitation studies on how reduce the fatigue post-stroke and its effects.

6. PLOS authors have the option to publish the peer review history of their article (what does this mean?). If published, this will include your full peer review and any attached files.

Reviewer #1: No

Reviewer #2: **Yes: **Marco Iosa

---

## [Author Response · Author response to Decision Letter 0]

9 Jul 2021

All responses are included in the rebuttal letter that was previously attached.

---

## [Editor Report · Decision Letter 1]

19 Jul 2021

Post-stroke fatigue: A factor associated with inability to return to work in patients <60 years - a 1-year follow-up.

PONE-D-21-12930R1

Dear Dr. Yang,

We’re pleased to inform you that your manuscript has been judged scientifically suitable for publication and will be formally accepted for publication once it meets all outstanding technical requirements.

Kind regards,

Simone Reppermund, PhD

Academic Editor

PLOS ONE

---

## [Editor Report · Acceptance letter]

26 Jul 2021

PONE-D-21-12930R1 

Post-stroke fatigue: A factor associated with inability to return to work in patients <60 years— a 1-year follow-up. 

Dear Dr. Yang:

I'm pleased to inform you that your manuscript has been deemed suitable for publication in PLOS ONE. Congratulations! Your manuscript is now with our production department. 

Kind regards, 

on behalf of

Dr. Simone Reppermund 

Academic Editor

PLOS ONE